# BRANCHED MULTI-TASK NETWORKS: DECIDING WHAT LAYERS TO SHARE

## ABSTRACT

In the context of multi-task learning, neural networks with branched architectures have often been employed to jointly tackle the tasks at hand. Such ramified networks typically start with a number of shared layers, after which different tasks branch out into their own sequence of layers. Understandably, as the number of possible network configurations is combinatorially large, deciding what layers to share and where to branch out becomes cumbersome. Prior works have either relied on ad hoc methods to determine the level of layer sharing, which is suboptimal, or utilized neural architecture search techniques to establish the network design, which is considerably expensive. In this paper, we go beyond these limitations and propose a principled approach to automatically construct branched multi-task networks, by leveraging the employed tasks' affinities. Given a specific budget, i.e. number of learnable parameters, the proposed approach generates architectures, in which shallow layers are task-agnostic, whereas deeper ones gradually grow more task-specific. Extensive experimental analysis across numerous, diverse multi-tasking datasets shows that, for a given budget, our method consistently yields networks with the highest performance, while for a certain performance threshold it requires the least amount of learnable parameters.

## 1 INTRODUCTION

Deep neural networks are usually trained to tackle different tasks in isolation. Humans, in contrast, are remarkably good at solving a multitude of tasks concurrently. Biological data processing appears to follow a multi-tasking strategy too; instead of separating tasks and solving them in isolation, different processes seem to share the same early processing layers in the brain – see e.g. V1 in macaques (Gur & Snodderly, 2007). Drawing inspiration from such observations, deep learning researchers began to develop multi-task networks with branched architectures.

As a whole, multi-task networks (Caruana, 1997) seek to improve generalization and processing efficiency through the joint learning of related tasks. Compared to the typical learning of separate deep neural networks for each of the individual tasks, multi-task networks come with several advantages. First, due to their inherent layer sharing (Kokkinos, 2017; Lu et al., 2017; Kendall et al., 2018; Guo et al., 2018; Liu et al., 2019), the resulting memory footprint is typically substantially lower. Second, as features in the shared layers do not need to be calculated repeatedly for the different tasks, the overall inference speed is often higher (Neven et al., 2017; Lu et al., 2017). Finally, multi-task networks may outperform their single-task counterparts (Kendall et al., 2018; Xu et al., 2018; Sener & Koltun, 2018; Maninis et al., 2019). Evidently, there is merit in utilizing multi-task networks.

When it comes to designing them, however, a significant challenge is to decide on the layers that need to be shared among tasks. Assuming a hard parameter sharing setting[1], the number of possible network configurations grows quickly with the number of tasks. As a result, a trial-and-error procedure to define the optimal architecture becomes unwieldy. Resorting to neural architecture

---

[1]In this setting, the input is first encoded through a stack of shared layers, after which tasks branch out into their own sequence of task-specific layers (Kokkinos, 2017; Lu et al., 2017; Kendall et al., 2018; Guo et al., 2018; Sener & Koltun, 2018). Alternatively, a set of task-specific networks can be used in conjunction with a feature sharing mechanism (Misra et al., 2016; Liu et al., 2019; Ruder et al., 2019). The latter approach is termed soft parameter sharing in the literature.

search (Elsken et al., 2019) techniques is not a viable option too, as in this case, the layer sharing has to be jointly optimized with the layers types, their connectivity, etc., rendering the problem considerably expensive. Instead, researchers have recently explored more viable alternatives, like routing (Rosenbaum et al., 2018), stochastic filter grouping (Bragman et al., 2019), and feature partitioning (Newell et al., 2019), which are, however, closer to the soft parameter sharing setting. Previous works on hard parameter sharing opted for the simple strategy of sharing the initial layers in the network, after which all tasks branch out simultaneously. The point at which the branching occurs is usually determined ad hoc (Kendall et al., 2018; Guo et al., 2018; Sener & Koltun, 2018). This situation hurts performance, as a suboptimal grouping of tasks can lead to the sharing of information between unrelated tasks, known as *negative transfer* (Zhao et al., 2018).

In this paper, we go beyond the aforementioned limitations and propose a novel approach to decide on the degree of layer sharing between tasks in order to eliminate the need for manual exploration. To this end, we base the layer sharing on measurable levels of *task affinity* or *task relatedness*: two tasks are strongly related, if their single task models rely on a similar set of features. Zamir et al. (2018) quantified this property by measuring the performance when solving a task using a variable sets of layers from a model pretrained on a different task. However, their approach is considerably expensive, as it scales quadratically with the number of tasks. Recently, Dwivedi & Roig (2019) proposed a more efficient alternative that uses representation similarity analysis (RSA) to obtain a measure of task affinity, by computing correlations between models pretrained on different tasks. Given a dataset and a number of tasks, our approach uses RSA to assess the task affinity at arbitrary locations in a neural network. The task affinity scores are then used to construct a branched multi-task network in a fully automated manner. In particular, our task clustering algorithm groups similar tasks together in common branches, and separates dissimilar tasks by assigning them to different branches, thereby reducing the negative transfer between tasks. Additionally, our method allows to trade network complexity against task similarity. We provide extensive empirical evaluation of our method, showing its superiority in terms of multi-task performance vs computational resources.

## 2 RELATED WORK

**Multi-task learning.** Multi-task learning (MTL) (Caruana, 1997; Ruder, 2017) is associated with the concept of jointly learning multiple tasks under a single model. This comes with several advantages, as described above. Early work on MTL often relied on sparsity constraints (Yuan & Lin, 2006; Argyriou et al., 2007; Lounici et al., 2009; Jalali et al., 2010; Liu et al., 2017) to select a small subset of features that could be shared among all tasks. However, this can lead to negative transfer when not all tasks are related to each other. A general solution to this problem is to cluster tasks based on prior knowledge about their similarity or relatedness (Evgeniou & Pontil, 2004; Abernethy et al., 2009; Agarwal et al., 2010; Zhou et al., 2011; Kumar & Daume III, 2012).

In the deep learning era, MTL models can typically be classified as utilizing soft or hard parameter sharing. In soft parameter sharing, each task is assigned its own set of parameters and a feature sharing mechanism handles the cross-task talk. Cross-stitch networks (Misra et al., 2016) softly share their features among tasks, by using a linear combination of the activations found in multiple single task networks. Sluice networks (Ruder et al., 2019) extend cross-stitch networks and allow to learn the selective sharing of layers, subspaces and skip connections. In a different vein, multi-task attention networks (Liu et al., 2019) use an attention mechanism to share a general feature pool amongst task-specific networks. In general, MTL networks using soft parameter sharing are limited in terms of scalability, as the size of the network tends to grow linearly with the number of tasks.

In hard parameter sharing, the parameter set is divided into shared and task-specific parameters. MTL models using hard parameter sharing are often based on a generic framework with a shared off-the-shelf encoder, followed by task-specific decoder networks (Neven et al., 2017; Kendall et al., 2018; Chen et al., 2018; Sener & Koltun, 2018). Multilinear relationship networks (Long et al., 2017) extend this framework by placing tensor normal priors on the parameter set of the fully connected layers. Guo et al. (2018) proposed the construction of a hierarchical network, which predicts increasingly difficult tasks at deeper layers. A limitation of the aforementioned approaches is that the branching points are determined ad hoc, which can easily lead to negative transfer if the predefined task groupings are suboptimal. In contrast, in our branched multi-task networks, the degree of layer sharing is automatically determined in a principled way, based on task affinities.

Our work bears some similarity to fully-adaptive feature sharing (Lu et al., 2017), which starts from a thin network where tasks initially share all layers, but the final one, and dynamically grows the model in a greedy layer-by-layer fashion. Task groupings, in this case, are decided on the probability of concurrently simple or difficult examples across tasks. Differently, (1) our method clusters tasks based on feature affinity scores, rather than example difficulty, which is arguably a better criterion; (2) the tree structure is determined offline using the precalculated affinities for the whole network, and not online in a greedy layer-by-layer fashion, which promotes task groupings that are optimal in a global, rather than local, sense; (3) our approach achieves significantly better results, especially on challenging datasets featuring numerous tasks, like Taskonomy.

**Neural architecture search.** Neural architecture search (NAS) (Elsken et al., 2019) aims to automate the construction of the network architecture. Different algorithms can be characterized based on their search space, search strategy or performance estimation strategy. Most existing works on NAS, however, are limited to task-specific models (Zoph & Le, 2017; Liu et al., 2018b; Pham et al., 2018; Liu et al., 2018a; Real et al., 2019). This is to be expected as when using NAS for MTL, layer sharing has to be jointly optimized with the layers types, their connectivity, etc., rendering the problem considerably expensive. To alleviate the heavy computation burden, a recent work (Liang et al., 2018) implemented an evolutionary architecture search for multi-task networks, while other researchers explored more viable alternatives, like routing (Rosenbaum et al., 2018), stochastic filter grouping (Bragman et al., 2019), and feature partitioning (Newell et al., 2019). In contrast to traditional NAS, the proposed methods do not build the architecture from scratch, but rather start from a predefined backbone network for which a layer sharing scheme is automatically determined.

**Transfer learning.** Transfer learning (Pan et al., 2010) makes use of the knowledge obtained when solving one task, and applies it to a different but related task. Our work is loosely related to transfer learning, as we use it to measure levels of task affinity. Zamir et al. (2018) provided a taxonomy for task transfer learning to quantify such relationships. However, their approach scales unfavorably w.r.t. the number of tasks, and we opted for a more efficient alternative proposed by Dwivedi & Roig (2019). The latter uses RSA to obtain a measure of task affinity, by computing correlations between models pretrained on different tasks. In our method, we use the performance metric from their work to compare the usefulness of different feature sets for solving a particular task.

**Loss weighting.** One of the known challenges of jointly learning multiple tasks is properly weighting the loss functions associated with the individual tasks. Early work (Kendall et al., 2018) used the homoscedastic uncertainty of each task to weigh the losses. Gradient normalization (Chen et al., 2018) balances the learning of tasks by dynamically adapting the gradient magnitudes in the network. Dynamic task prioritization (Guo et al., 2018) prioritizes the learning of difficult tasks. Zhao et al. (2018) observed that two competing tasks can cause the destructive interference of the gradient, and proposed a modulation module to alleviate this problem. Sener & Koltun (2018) cast multi-task learning as a multi-objective optimization problem, with the overall objective of finding a Pareto optimal solution. Note that, addressing the loss weighting issue in MTL is out of the scope of this work. In fact, all our experiments are based on a simple uniform loss weighing scheme.

## 3 METHOD

In this paper, we aim to jointly solve $N$ different tasks $\mathcal{T} = \{t_1, \ldots, t_N\}$ given a computational budget $\mathcal{C}$, i.e. number of parameters. Consider a backbone architecture: an encoder, consisting of a sequence of shared layers or blocks $f_l$, followed by a decoder with a few task-specific layers. We assume an appropriate structure for layer sharing to take the shape of a tree. In particular, the first layers are shared by all tasks, while later layers gradually split off as they show more task-specific behavior. The proposed method aims to find an effective task grouping for the sharable layers $f_l$ of the encoder, i.e. grouping related tasks together in the same branches of the tree. When two tasks are strongly related, we expect their single-task models to rely on a similar feature set (Zamir et al., 2018). Based on this viewpoint, the proposed method derives a task affinity score at various locations in the sharable encoder. The resulting task affinity scores are used for the automated construction of a branched multi-task network that fits the computational budget $\mathcal{C}$. Fig. 1 illustrates our pipeline, while Algorithm 1 summarizes the whole procedure.

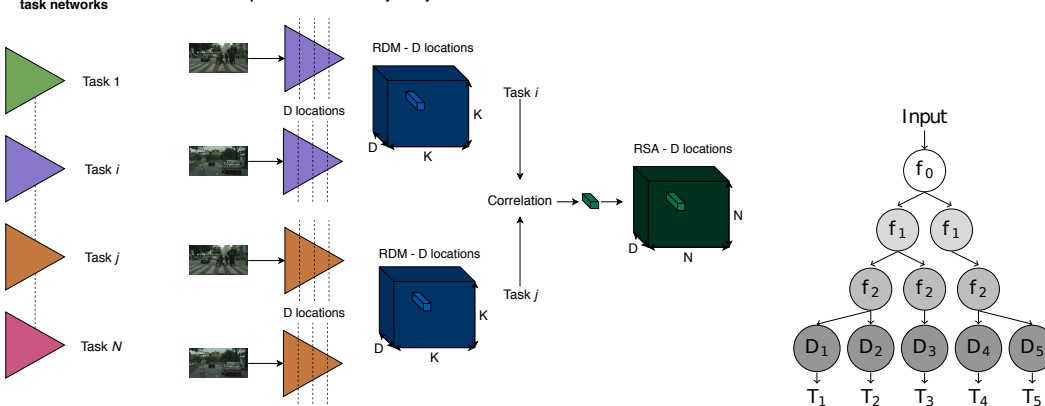

(a) Pipeline overview. (left) We train a single-task model for every task $t$ in $\mathcal{T}$. (middle) We use RSA to measure the task affinity at $D$ predefined locations in the sharable encoder. In particular, we calculate the representation dissimilarity matrices (RDM) for the features at $D$ locations using $K$ images, which gives a $D \times K \times K$ tensor per task. (right) The affinity tensor **A** is found by calculating the correlation between the RDM matrices, which results in a three-dimensional tensor of size $D \times N \times N$, with $N$ the number of tasks.

(b) Our pipeline's output is a branched multi-task network, similar to how NAS techniques output sample architectures. An example branched multi-task network is visualized here.

Figure 1: The proposed method: (a) calculate task affinities at various locations in the sharable encoder; (b) construct a branched multi-task network based on the computed affinities.

---

**Algorithm 1** Branched Multi-Task Networks - Task clustering

1: **Input:** Tasks $\mathcal{T}$, $K$ images $\mathcal{I}$, a sharable encoder $E$ with $D$ locations where we can branch, a set of task specific decoders $D_t$ and a computational budget $\mathcal{C}$.
2: **for** $t$ in $\mathcal{T}$ **do**
3:     Train the encoder $E$ and task-specific decoder $D_t$ for task $t$.
4:     $\mathbf{RDM}^t \leftarrow RDM(E, D, \mathcal{I})$                             $\triangleright$ RDM for task $t$.
5: **end for**
6: $A_{d,i,j} \leftarrow r_s \left( \text{triu} \left( \mathbf{RDM}^{t_i}_{d,:,:} \right), \text{triu} \left( \mathbf{RDM}^{t_j}_{d,:,:} \right) \right)$ **for** $t_i, t_j$ in $\mathcal{T}$ and $d$ in locations    $\triangleright$ Task affinity
7: $\mathbf{D} = 1 - \mathbf{A}$                                              $\triangleright$ Task dissimilarity
8: **Return:** Task-grouping with minimal task dissimilarity that fits within $\mathcal{C}$

---

### 3.1 CALCULATE TASK AFFINITY SCORES

As mentioned, we rely on RSA to measure task affinity scores. This technique has been widely adopted in the field of neuroscience to draw comparisons between behavioral models and brain activity. Inspired by how Dwivedi & Roig (2019) applied RSA to select tasks for transfer learning, we use the technique to assess the task affinity at predefined locations in the sharable encoder. Consequently, using the measured levels of task affinity, tasks are assigned in the same or different branches of a branched multi-task network, subject to the computational budget $\mathcal{C}$.

The procedure to calculate the task affinity scores is the following. As a first step, we train a single-task model for each task $t_i \in \mathcal{T}$. The single-task models use an identical encoder $E$ - made of all sharable layers $f_l$ - followed by a task-specific decoder $D_{t_i}$. The decoder contains only task-specific operations and is assumed to be significantly smaller in size compared to the encoder. As an example, consider jointly solving a classification and a dense prediction task. Some fully connected layers followed by a softmax operation are typically needed for the classification task, while an additional decoding step with some upscaling operations is required for the dense prediction task. Of course, the appropriate loss functions are applied in each case. Such operations are part of the task-specific decoder $D_{t_i}$. The different single-task networks are trained under the same conditions.

At the second step, we choose $D$ locations in the sharable encoder $E$ where we calculate a two-dimensional task affinity matrix of size $N \times N$. When concatenated, this results in a three-dimensional tensor **A** of size $D \times N \times N$ that holds the task affinities at the selected locations. To

calculate these task affinities, we have to compare the representation dissimilarity matrices (RDM) of the single-task networks - trained in the previous step - at the specified $D$ locations. To do this, a held-out subset of $K$ images is required. The latter images serve to compare the dissimilarity of their feature representations in the single-task networks for every pair of images. Specifically, for every task $t_i$, we characterize these learned feature representations at the selected locations by filling a tensor of size $D \times K \times K$. This tensor contains the dissimilarity scores $1 - \rho$ between feature representations, with $\rho$ the Pearson correlation coefficient. Specifically, $\mathbf{RDM}_{d,i,j}$ is found by calculating the dissimilarity score between the features at location $d$ for image $i$ and $j$.

For a specific location $d$ in the network, the computed RDMs are symmetrical, with a diagonal of zeros. For every such location, we measure the similarity between the upper or lower triangular part of the RDMs belonging to the different single-task networks. We use the Spearman's correlation coefficient $r_s$ to measure similarity. When repeated for every pair of tasks, at a specific location $d$, the result is a symmetrical matrix of size $N \times N$, with a diagonal of ones. Concatenating over the $D$ locations in the sharable encoder, we end up with the desired task affinity tensor of size $D \times N \times N$. Note that, in contrast to prior work (Lu et al., 2017), the described method focuses on the features used to solve the single tasks, rather than the examples and how easy or hard they are across tasks, which is arguably a better measure of task affinity.

### 3.2 Construct a branched multi-task network

Given a computational budget $\mathcal{C}$, we need to derive how the layers (or blocks) $f_l$ in the sharable encoder $E$ should be shared among the tasks in $\mathcal{T}$. Each layer $f_l \in E$ is represented as a node in the tree, i.e. the root node contains the first layer $f_0$, and nodes at depth $l$ contain layer(s) $f_l$. The granularity of the layers $f_l$ corresponds to the intervals at which we measure the task affinity in the sharable part of the model, i.e. the $D$ locations. When the encoder is split into $b_l$ branches at depth $l$, this is equivalent to a node at depth $l$ having $b_l$ children. The task-specific decoders $D_t$ can be found in the leaves of the tree. Fig. 1b shows an example of such a tree using the aforementioned notation. Each node is responsible for solving a unique subset of tasks.

The branched multi-task network is built with the intention to separate dissimilar tasks by assigning them to separate branches. To this end, we define the dissimilarity score between two tasks $t_i$ and $t_j$ at location $d$ as $1 - \mathbf{A}_{d,i,j}$, with $\mathbf{A}$ the task affinity tensor[2]. The branched multi-task network is found by minimizing the sum of the task dissimilarity scores at every location in the sharable encoder. In contrast to prior work (Lu et al., 2017), the task affinity (and dissimilarity) scores are calculated a priori. This allows us to determine the task clustering offline. Since the number of tasks is finite, we can enumerate all possible trees that fall within the given computational budget $\mathcal{C}$. Finally, we select the tree that minimizes the task dissimilarity score. The task dissimilarity score of a tree is defined as $C_{cluster} = \sum_l C_{cluster}^l$, where $C_{cluster}^l$ is found by averaging the maximum distance between the dissimilarity scores of the elements in every cluster. The use of the maximum distance encourages the separation of dissimilar tasks. By taking into account the clustering cost at all depths, the procedure can find a task grouping that is considered optimal in a global sense. This is in contrast to the greedy approach in (Lu et al., 2017), which only minimizes the task dissimilarity locally, i.e. at isolated locations in the network.

An exhaustive search becomes intractable when the number of tasks is extremely large. For such cases, we propose to derive the tree in a top-down manner, starting at the most outer layer. At every step $l$, we can perform spectral clustering for each possible number of groups $m$ where $1 \leq m \leq b_{l+1}$, with $b_{l+1}$ the number of branches at layer $l + 1$. Before proceeding to the next step, we select the top-$n$ task groupings with minimal cost. This constrains the number of possible groupings at the next layer. When we proceed to cluster the tasks at the next layer, we select the top-$n$ groupings from the ones that are still eligible to be constructed. This beam search is used in CelebA experiments.

## 4 Experiments

In this section, we quantitatively and qualitatively evaluate the proposed method on a number of diverse multi-tasking datasets, that range from real to semi-real data, from few to many tasks, from dense prediction to classification tasks, and so on.

---

[2]This is not to be confused with the dissimilarity score used to calculate the RDM elements $\mathbf{RDM}_{d,i,j}$.

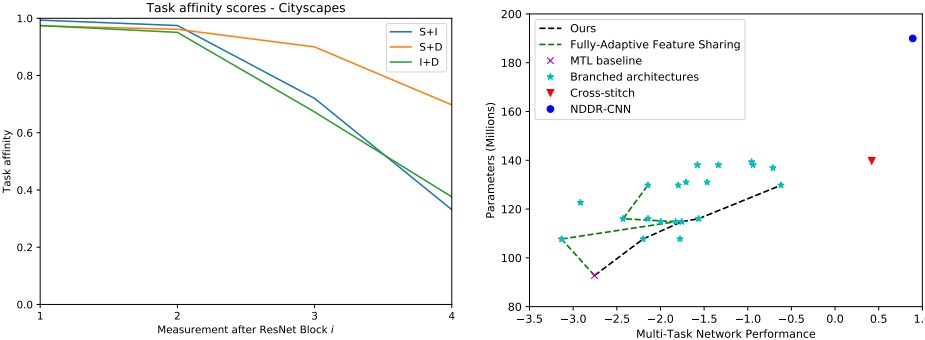

(a) Task affinity after each ResNet-50 block.

(b) Number of parameters versus multi-task performance on the Cityscapes validation set.

Figure 2: Qualitative analysis on the Cityscapes dataset.

Table 1: Quantitative analysis on the Cityscapes validation set.

| Method | S (mIoU) | I (px) | D (px) | Parameters (M) | MTL performance (%) |
|---|---|---|---|---|---|
| Single task | 65.17 | 11.70 | 2.57 | 138 | 0.00 |
| MTL baseline | 61.51 | 11.80 | 2.66 | 92 | -3.28 |
| Cross-stitch | 65.14 | 11.63 | 2.55 | 140 | 0.42 |
| NDDR-CNN | 65.64 | 11.61 | 2.54 | 190 | 0.89 |
| Ours - 1 | 62.14 | 11.74 | 2.66 | 107 | -2.81 |
| Ours - 2 | 62.67 | 11.67 | 2.62 | 114 | -1.90 |
| Ours - 3 | 64.11 | 11.62 | 2.62 | 116 | -1.00 |

## 4.1 CITYSCAPES

**Dataset.** The Cityscapes dataset (Cordts et al., 2016) considers the challenging scenario of urban scene understanding. The train, validation and test set contain respectively 2975, 500 and 1525 real images, taken by driving a car in Central European cities. It considers a few dense prediction tasks: semantic segmentation (S), instance segmentation (I) and monocular depth estimation (D). As in prior works (Kendall et al., 2018; Sener & Koltun, 2018), we use a ResNet-50 encoder with dilated convolutions, followed by a Pyramid Spatial Pooling (PSP) (He et al., 2015) decoder. Every input image is rescaled to 512 x 256 pixels. We reuse the approach from Kendall et al. (2018) for the instance segmentation task, i.e. we consider the proxy task of regressing each pixel to the center of the instance it belongs to. We obtained all results after a grid search on the hyperparameter space, to ensure a fair comparison across the compared approaches. For more details please visit Appendix A.

**Results.** We measure the task affinity after every block (1 to 4) in the ResNet-50 model (see Fig. 2a). The task affinity decreases in the deeper layers of the model, due to the features becoming more task-specific. We compare the performance of the task groupings generated by our method with those by other approaches. As in (Maninis et al., 2019), the performance of a multi-task model $m$ is defined as the average per-task performance drop/increase w.r.t. a single-task baseline $b$.

We trained all possible task groupings that can be derived from branching the model in the last three ResNet blocks. Fig. 2b visualizes performance vs number of parameters for the trained architectures. Depending on the available computational budget $\mathcal{C}$, our method generates a specific task grouping. We visualize these generated groupings as a path in Fig. 2b, when gradually increasing the computational budget $\mathcal{C}$. Similarly, we consider the task groupings when branching the model based on the task affinity measure proposed by Lu et al. (2017). We find that, in comparison, the task groupings devised by our method achieve higher performance within a given computational budget $\mathcal{C}$. Furthermore, in the majority of cases, for a fixed budget $\mathcal{C}$ the proposed method is capable of selecting the best performing task grouping w.r.t. performance vs parameters metric.

Table 2: Quantitative analysis on the tiny Taskonomy test set. The results for edge (E) and keypoints (K) detection were multiplied by a factor of 100 for better readability. The FA models refer to generating the task groupings with the task affinity technique proposed by Lu et al. (2017).

| Method | D(L1) | S(mIoU) | C(top-5) | E(L1) | K(L1) | Params(M) | MTL Performance (%) |
|---|---|---|---|---|---|---|---|
| Single task | 0.60 | 43.45 | 66.02 | 0.99 | 0.23 | 224 | 0.0 |
| MTL baseline | 0.75 | 47.82 | 55.99 | 1.37 | 0.34 | 130 | -22.50 |
| Cross-stitch | 0.61 | 43.99 | 58.24 | 1.35 | 0.50 | 224 | -32.29 |
| NDDR-CNN | 0.66 | 45.94 | 64.48 | 1.05 | 0.45 | 258 | -21.02 |
| FA-1 | 0.74 | 46.14 | 62.73 | 1.30 | 0.39 | 174 | -24.5 |
| FA-2 | 0.80 | 39.93 | 62.36 | 1.68 | 0.52 | 188 | -48.32 |
| FA-3 | 0.74 | 46.09 | 64.93 | 1.05 | 0.27 | 196 | -8.48 |
| Ours-1 | 0.76 | 47.62 | 63.28 | 1.12 | 0.29 | 174 | -11.88 |
| Ours-2 | 0.74 | 47.96 | 63.61 | 0.96 | 0.35 | 188 | -12.66 |
| Ours-3 | 0.74 | 47.88 | 64.54 | 0.94 | 0.26 | 196 | -4.93 |

We also compare our branched multi-task networks with cross-stitch networks (Misra et al., 2016) and NDDR-CNNs (Gao et al., 2019) in Table 1[3]. While the latter give higher multi-task performance, attributed to their computationally expensive soft parameter sharing setting, our branched multi-task networks can strike a better trade-off between the performance and number of parameters. In particular, Fig. 2b shows that we can effectively sample architectures which lie between the extremes of a baseline multi-task model and a cross-stitch or NDDR-CNN architecture. It is worth noting that soft parameter sharing does not scale when the number of tasks increases greatly.

## 4.2 TASKONOMY

**Dataset.** The Taskonomy dataset (Zamir et al., 2018) contains semi-real images of indoor scenes, annotated for 26 (dense preciction, classification, etc.) tasks. Out of the available tasks, we select scene categorization (C), semantic segmentation (S), edge detection (E), monocular depth estimation (D) and keypoint detection (K). The task dictionary was selected to be as diverse as possible, while still keeping the total number of tasks reasonable for all computations. We use the tiny split of the dataset, containing 275k train, 52k validation and 54k test images. We reuse the architecture and training setup from Zamir et al. (2018): the encoder is based on ResNet-50; a 15-layer fully-convolutional decoder is used for the pixel-to-pixel prediction tasks. Appendix B contains a more detailed description on the training setup of the Taskonomy experiments.

**Results.** The task affinity is again measured after every ResNet block. Since the number of tasks increased to five, it is very expensive to train all task groupings exhaustively, as done above. Instead, we limit ourselves to three architectures that are generated when gradually increasing the parameter budget. As before, we compare our task groupings against the method from Lu et al. (2017). The numerical results can be found in Table 2. The task groupings themselves are shown in Appendix B.

The effect of the employed task grouping technique can be seen from comparing the performance of our models against the corresponding FA models, generated by (Lu et al., 2017). The latter are consistently outperformed by our models. Compared to the results on Cityscapes (Fig. 2b), we find that the multi-task performance is much more susceptible to the employed task groupings, possibly due to negative transfer. Furthermore, we observe that cross-stitch networks and NDDR-CNNS can not handle the larger, more diverse task dictionary: the performance decreases when using these models, while the number of parameters increases. This is in contrast to our branched multi-task networks, which seem to handle the diverse set of tasks rather positively. As opposed to (Zamir et al., 2018), but in accordance with (Maninis et al., 2019), we show that it is possible to solve many heterogeneous tasks simultaneously when the negative transfer is limited, by separating dissimilar tasks from each other in our case. In fact, our approach is the first to show such consistent performance across different multi-tasking scenarios and datasets. Existing approaches seem to be tailored for particular cases, e.g. few/correlated tasks, synthetic-like data, binary classification only tasks, etc., whereas we show stable performance across the board of different experimental setups.

---

[3]We included MTAN (Liu et al., 2019) in our experiments too, but we were unable to solve all three tasks on Cityscapes (S, I, D) simultaneously when using their proposed architecture.

Table 3: Quantitative analysis on the CelebA test set. **(bold)** The Ours-Thin-32 architecture is found by optimizing the task clustering for the parameter budget that is used in the Branch-32-2.0 model. *(italic)* The Ours-Thin-64 architecture is found by optimizing the task clustering for the parameter budget that is used in the GNAS-Shallow-Wide model.

| Method | Accuracy (%) | Parameters (Millions) |
|---|---|---|
| LNet+ANet (Wang et al., 2016) | 87 | - |
| Walk and Learn (Wang et al., 2016) | 88 | - |
| MOON (Rudd et al., 2016) | 90.94 | 119.73 |
| Independent Group (Hand & Chellappa, 2017) | 91.06 | - |
| MCNN (Hand & Chellappa, 2017) | 91.26 | - |
| MCNN-AUX (Hand & Chellappa, 2017) | 91.29 | - |
| VGG-16 Baseline (Lu et al., 2017) | 91.44 | 134.41 |
| **Branch-32-2.0** (Lu et al., 2017) | **90.79** | **2.09** |
| GNAS-Shallow-Thin (Hand & Chellappa, 2017) | 91.30 | 1.57 |
| *GNAS-Shallow-Wide* (Hand & Chellappa, 2017) | *91.63* | *7.73* |
| GNAS-Deep-Thin (Hand & Chellappa, 2017) | 90.90 | 1.47 |
| GNAS-Deep-Wide (Hand & Chellappa, 2017) | 91.36 | 6.41 |
| ResNet-18 (Uniform weighing) (Sener & Koltun, 2018) | 90.38 | 11.2 |
| ResNet-18 (MGDA-UB) (Sener & Koltun, 2018) | 91.75 | 11.2 |
| **Ours-Thin-32** | **91.46** | **2.20** |
| *Ours-Thin-64* | *91.73* | *7.73* |

## 4.3 CELEBA

**Dataset.** The CelebA dataset (Liu et al., 2015) contains over 200k real images of celebrities, labeled with 40 facial attribute categories. The training, validation and test set contain 160k, 20k and 20k images respectively. We treat the prediction of each facial attribute as a single binary classification task, as in (Lu et al., 2017; Sener & Koltun, 2018; Huang et al., 2018). To ensure a fair comparison: we reuse the thin-$\omega$ model from Lu et al. (2017) in our experiments on CelebA; the parameter budget $\mathcal{C}$ is set for the model to have the same amount of parameters as prior work. As mentioned in Sec. 3.2, we use the beam search adaptation of our optimization procedure due to the very large number of tasks. We set $n = 10$, with the top-$n$ being the number of groupings to keep at every layer during the optimization. Note that, the final result remains unchanged when applying small changes to $n$. Appendix C provides more details on the training setup.

**Results.** Table 3 shows the results on the CelebA test set. The task groupings themselves are visualized in Appendix C. Our branched multi-task networks outperform earlier works (Lu et al., 2017; Huang et al., 2018) when using a similar amount of parameters. Since our Thin-32 model only differs from the model in (Lu et al., 2017) on the employed task grouping technique, we can conclude that the proposed method devises more effective task groupings for the attribute classification tasks on CelebA. Furthermore, our Thin-32 model performs on par with the VGG-16 baseline, while using 64 times less parameters. We also compare our results with the ResNet-18 model from Sener & Koltun (2018). Our Thin-64 models performs 1.35% better than the ResNet-18 model when trained with a uniform loss weighing scheme. More noticeably, our Thin-64 model performs on par with the state-of-the-art ResNet-18 model that was trained with the loss weighing scheme from Sener & Koltun (2018), while at the same time using 31% less parameters (11.2 vs 7.7 M).

## 5 CONCLUSION

In this paper, we introduced a principled approach to automatically construct branched multi-task networks for a given computational budget. To this end, we leverage the employed tasks' affinities as a quantifiable measure for layer sharing. The proposed approach can be seen as an abstraction of NAS for MTL, where only layer sharing is optimized, without having to jointly optimize the layers types, their connectivity, etc., as done in traditional NAS, which would render the problem considerably expensive. Extensive experimental analysis shows that our method outperforms existing ones w.r.t. the important metric of multi-tasking performance vs number of parameters, while at the same time showing consistent results across a diverse set of multi-tasking scenarios and datasets.

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

## A    CITYSCAPES

The encoder is a ResNet-50 model with dilated convolutions (Yu & Koltun, 2015), pre-trained on ImageNet. We use a PSP module (He et al., 2015) for the task-specific decoders. Every input image is rescaled to 512 x 256 pixels. We upsample the output of the PSP decoders back to the input resolution during training. The outputs are upsampled to 2048 x 1024 pixels during testing. The semantic segmentation task is learned with a weighted pixel-wise cross-entropy loss. We reuse the approach from Kendall et al. (2018) for the instance segmentation task, i.e. we consider the proxy task of regressing each pixel to the center of the instance it belongs to. The depth estimation task is learned using an L1 loss. The losses are normalized to avoid having the loss of one task overwhelm the others during training. The hyperparameters were optimized with a grid search procedure to ensure a fair comparison across all compared approaches.

**Single-task models**    We tested batches of size 4, 6 and 8, poly learning rate decay vs step learning rate decay with decay factor 10 and step size 30 epochs, and Adam (initial learning rates 2e-4, 1e-4, 5e-5, 1e-5) vs stochastic gradient descent with momentum 0.9 (initial learning rates 5e-2, 1e-2, 5e-3, 1e-3). This accounts for 48 hyperparameter settings in total. We repeated this procedure for every single task (semantic segmentation, instance segmentation and monocular depth estimation).

**Baseline multi-task network**    We train with the same set of hyperparameters as before, i.e. 48 settings in total. We calculate the multi-task performance in accordance with Maninis et al. (2019). In particular, the multi-task performance of a model $m$ is measured as the average per-task performance increase/drop w.r.t. the single task models $b$:

$$\Delta_m = \frac{1}{T} \sum_{i=1}^{T} (-1)^{l_i} \left( M_{m,i} - M_{b,i} \right) / M_{b,i}, \tag{1}$$

where $l_i = 1$ if a lower value means better for measure $M_i$ of task $i$, and 0 otherwise.

**Branched multi-task network**    We reuse the hyperparameter setting with the best result for the baseline multi-task network. The branched multi-task architectures from Table 1 are shown in Fig. 3.

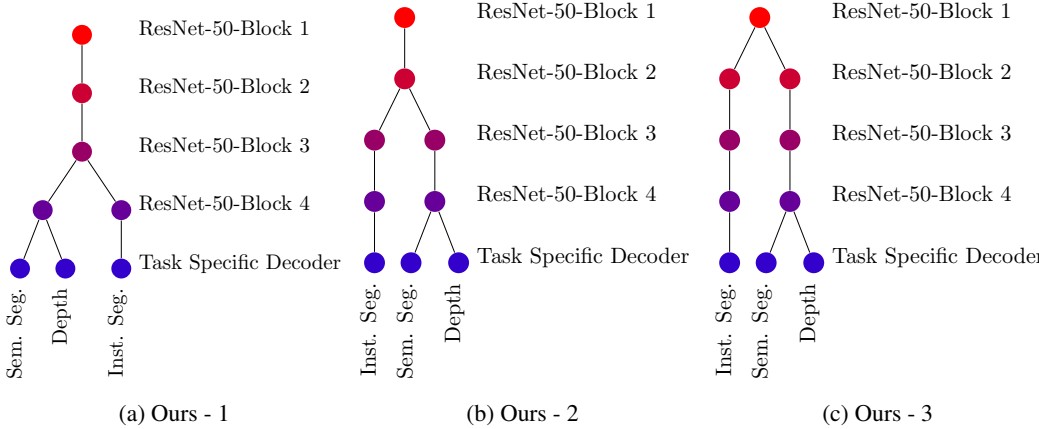

Figure 3: Branched multi-task networks on Cityscapes that were generated by our method.

**Cross-stitch networks / NDDR-CNN**    We insert a cross-stitch/NDDR unit after every ResNet block. We also tried to leave out the cross-stitch/NDDR unit after the final ResNet block, but this decreased performance. We tested two different initialization schemes for the weights in the cross-stitch/NDDR units, i.e. $\alpha = 0.8$, $\beta = 0.1$ and $\alpha = 0.9$, $\beta = 0.05$. The model weights were initialized from the set of the best single-task models above. We found that the Adam optimizer broke the initialization and refrained from using it. The best results were obtained with stochastic gradient descent with initial learning rate 1e-3 and momentum 0.9. As also done in (Misra et al., 2016; Gao et al., 2019), we set the weights of these units to have a learning rate that is 100 times higher than the base learning rate.

**MTAN**  We tried re-implementing the MTAN model (Liu et al., 2019) using a ResNet-50 back-bone. The architecture was based on the Wide-ResNet architecture that is used in the original paper. After extensive hyperparameter tuning, we were unable to get a meaningful result on the Cityscapes dataset when trying to solve all three tasks jointly. Note that, the authors have only shown results in their paper when training semantic segmentation and monocular depth estimation.

## B  TASKONOMY

We reuse the setup from Zamir et al. (2018). All input images were rescaled to 256 x 256 pixels. We use a ResNet-50 encoder and replace the last stride 2 convolution by a stride 1 convolution. A 15-layer fully-convolutional decoder is used for the pixel-to-pixel prediction tasks. The decoder is composed of five convolutional layers followed by alternating convolutional and transposed convolutional layers. We use ReLU as non-linearity. Batch normalization is included in every layer except for the output layer. We use Kaiming He's initialization for both encoder and decoder. We use an L1 loss for the depth (D), edge detection (E) and keypoint detection (K) tasks. The scene categorization task is learned with a KL-divergence loss. We report performance on the scene categorization task by measuring the overlap in top-5 classes between the predictions and ground truth.

The multi-task models were optimized with task weights $w_s = 1, w_d = 1, w_k = 10, w_e = 10$ and $w_c = 1$. Notice that the heatmaps were linearly rescaled to lie between 0 and 1. During training we normalize the depth map by the standard deviation.

**Single-task models**  We use an Adam optimizer with initial learning rate 1e-4. The learning rate is decayed by a factor of 10 after 80000 iterations. We train the model for 120000 iterations. The batch size is set to 32. No additional data augmentation is applied. The weight decay term is set to 1e-4.

**Baseline multi-task model**  We use the same optimization procedure as for the single-task models. The multi-task performance is calculated using Eq. 1.

**Branched multi-task models**  We use the same optimization procedure as for the single-task models. The architectures that were generated by our method are shown in Fig. 4. Fig. 5 shows the architectures that are found when using the task grouping method from Lu et al. (2017). We show some of the predictions made by our third branched multi-task network in figure 6 for the purpose of qualitative evaluation.

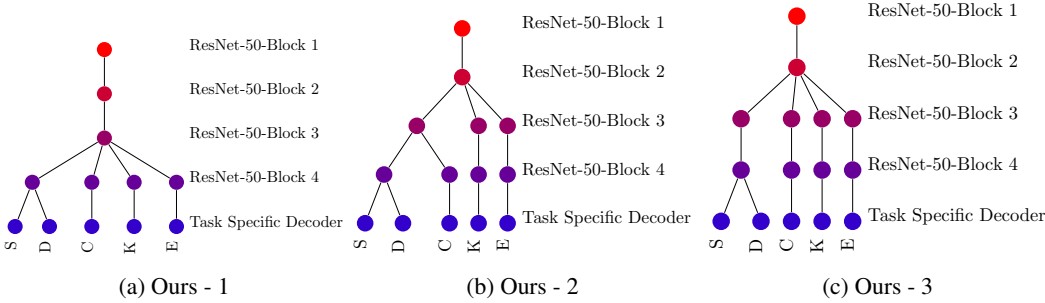

Figure 4: Task groupings generated by our method. The numerical results can be found in Table 2.

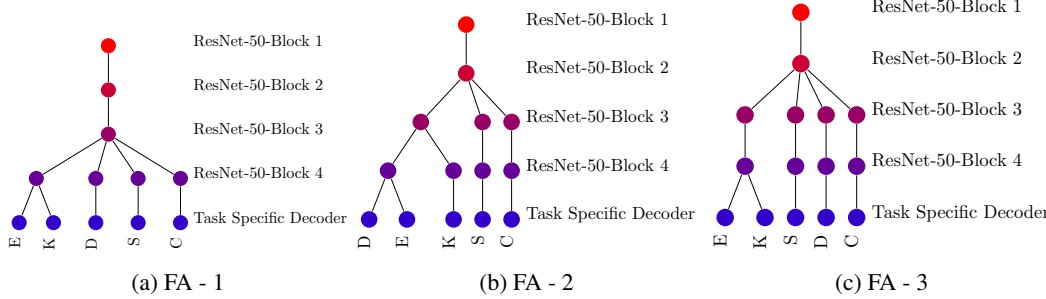

Figure 5: Task groupings generated using the method from Lu et al. (2017). The numerical results can be found in Table 2.

**Cross-stitch networks / NDDR-CNN** We reuse the hyperparameter settings that were found optimal on Cityscapes. Note that, these are in agreement with what the authors reported in their original papers. The weights of the cross-stitch/NDDR units were initialized with $\alpha = 0.8$ and $\beta = 0.05$.

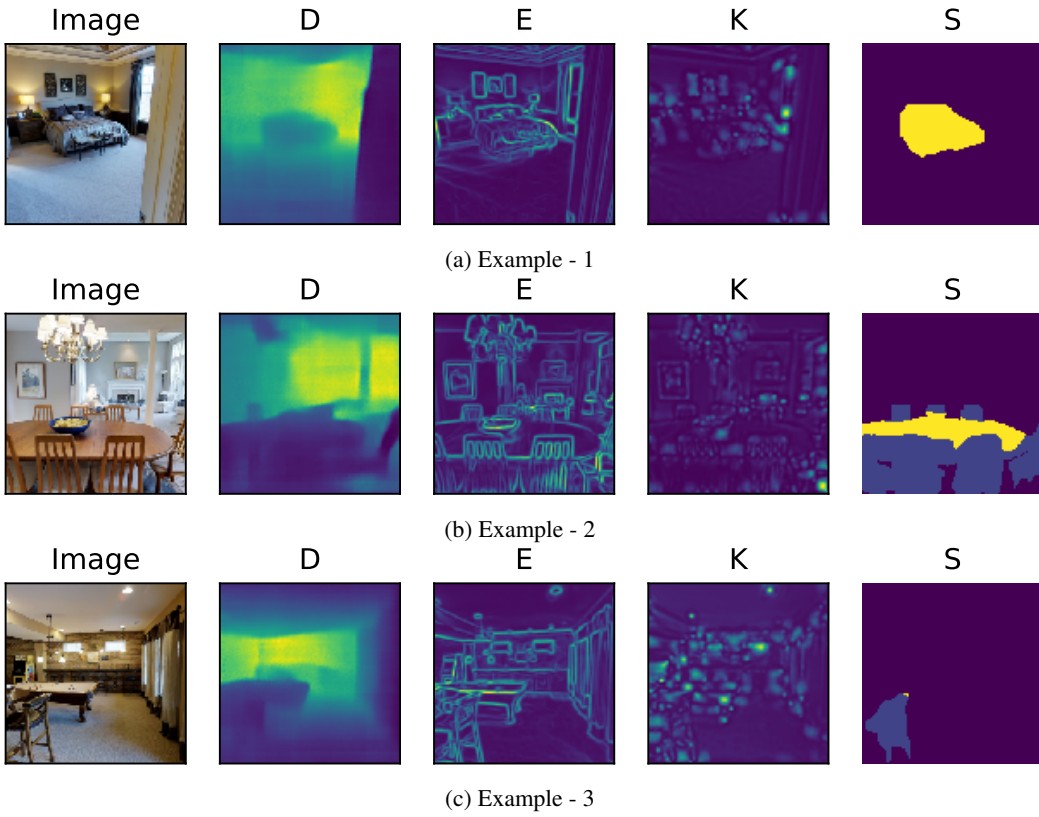

Figure 6: Predictions made by our branched multi-task network on images from the Taskonomy test set.

## C    CELEBA

We reuse the thin-$\omega$ model from Lu et al. (2017). The CNN architecture is based on the VGG-16 model (Simonyan & Zisserman, 2015). The number of convolutional features is set to the minimum between $\omega$ and the width of the corresponding layer in the VGG-16 model. The fully connected layers contain $2 \cdot \omega$ features. We train the branched multi-task network using stochastic gradient descent with momentum $0.9$ and initial learning rate $0.05$. We use batches of size 32 and weight

decay 0.0001. The model is trained for 120000 iterations and the learning rate divided by 10 every 40000 iterations. The loss function is a sigmoid cross-entropy loss with uniform weighing scheme.

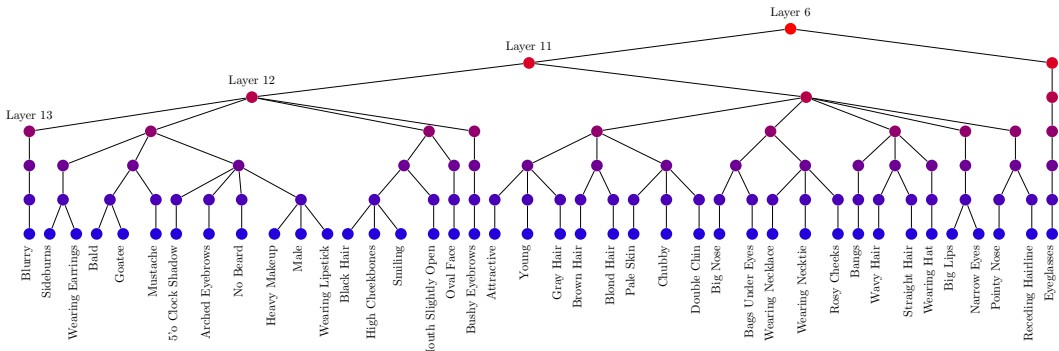

Figure 7: Grouping of 40 person attribute classification tasks on CelebA in a thin VGG-16 architecture.

