# OpenReview forum: "Branched Multi-Task Networks: Deciding What Layers To Share"
_ICLR.cc/2020/Conference — Reject_

### Official Review · AnonReviewer3 · 2019-10-22
**Official Blind Review #3**

**Rating:** 3

**Review:**

This paper proposes a novel soft parameter sharing Multi-task Learning framework based on a tree-like structure. The idea is interesting.

However, the technique details, the experimental results and the analysis are not as attractive as the idea. The proposed method is a simple combination of existing works without any creative improvement. Furthermore, comparing with the MTL baseline, the experimental performance of proposed method does not get obvious improvement while the computation cost increasing significantly.  Besides, there is not enough analysis about the idea this paper proposed. The intuition that more similar tasks share more parameters probably cannot always ensure the improvement of MTL.


**Experience Assessment:**

I have read many papers in this area.

**Review Assessment: Checking Correctness Of Derivations And Theory:**

I assessed the sensibility of the derivations and theory.

**Review Assessment: Checking Correctness Of Experiments:**

I carefully checked the experiments.

**Review Assessment: Thoroughness In Paper Reading:**

I read the paper at least twice and used my best judgement in assessing the paper.

---

> ### Author Response · Authors · 2019-11-12
> **Response to reviewer 3**
>
> We thank Reviewer 3 for his/her comments, however we would like to point out an important issue here. Having continuously served as reviewers ourselves, generic reviews, like this one, are against the review guidelines handed to the reviewers, and do not help promote future improvements in the paper based on analytic feedback from the reviewers.
>
> Having said that, let us elaborate on more specific points in the review. According to Reviewer 3, ‘the technique details, the experimental results and the analysis are not as attractive as the idea’. Can you be more specific here? What extra technique details would you think are currently lacking from the paper and should be added? Why are experimental results and analysis not supporting the idea? In fact, this is in contrast to what the other reviewers claim in the positive points of their reviews. Indeed, we performed extensive experimental evaluation to show the effectiveness of our approach. It was shown for two different backbones (Resnet, VGG) on three different datasets (CelebA, Cityscapes, Taskonomy) that our method improves over the multi-task baseline. On CelebA, our comparison in table 3 shows that our model can perform on par with the state-of-the-art (Sener & Koltun, 2018) while using less parameters (-31%). On Cityscapes it was shown that for the selected anchor points, our method leads to the best task groupings for a given amount of parameters. When considering five tasks on Taskonomy, our comparison with prior work (MTL baseline; Lu et al., 2017; Misra et al.,2016; Gao et al.,2019) showed that our method is best suited to handle such a diverse set of tasks. Specifically, our task groupings outperform the ones found by the method from Lu et al., 2017. Cross-stitch networks (Misra et. al,. 2016) and NDDR-CNNs (Gao et al., 2019) increase the number of parameters compared to a set of single task networks, while hurting the performance. Also on Taskonomy we show that our model improves over the MTL baseline by separating dissimilar tasks from each other. In general, we have put a lot of effort to perform a fair comparison between different architectures (see the extensive description of the experiments in the supplementary materials) and do not see the point-of-view in the review of Reviewer 3 regarding the results. We would again invite him/her to make more specific points and we would be more than happy to answer them.

---

> > ### Comment · AnonReviewer3 · 2019-11-13
> > **Specific Points**
> >
> > 1.	The paper announces that it presents a novel multi-task learning framework without any assumption of application. However, in existing RSA (Representation Similarity Analysis) based researches, it is only used in computer vision tasks. The experiments of the paper are all computer vison tasks. Multi-task learning is technology which can be used in various application (e.g. text classification, survival analysis with DNA data).  The authors should provide more experiments on other applications.
> >
> > 2.	In the end of section 3.1, the paper presents : Note that, in contrast to prior work (Lu et al., 2017), the described method focuses on the features used to solve the single tasks, rather than the examples and how easy or hard they are across tasks, which is arguably a better measure of task affinity. There is no theoretical evidence showing that measuring relatedness in feature space is better than in sample space. The authors can refer it in the paper [1][2], which present the uniform convergence properties of MTL and multi-task representation learning. Besides, the authors have not provided any experimental support. The note is not serious.
> >
> > 3.	The paper proposes to derive the tree in a top-down manner. To compare with exhaustive search, the specific computational complexity should be given.
> >
> > 4.	In the experiments, the authors present that they have done extensive experimental evaluation. However, the most basic comparison condition is not given. How many epochs has been done in the experiments? Whether the accuracy is the highest accuracy or average accuracy achieved by models generated in the last N epochs. If it is the highest accuracy, I think several experiment results are not evidence enough. For example, in table 3, the performance of GNAS-Shallow-Thin is quite close to the proposed method. As we know, during training, the accuracy of models generated different epoch is fluctuated. The highest accuracy may be just achieved by chance.
> >
> > 5.	In the experiments, except the Branched MTL, the networks structures are deeper than the baselines. This paper has not explained whether the performance improvement comes from the extra layers. More comparison experiments should be given.
> >
> > [1] Baxter J. A model of inductive bias learning[J]. Journal of artificial intelligence research, 2000, 12: 149-198.
> > [2] Maurer A, Pontil M, Romera-Paredes B. The benefit of multitask representation learning[J]. The Journal of Machine Learning Research, 2016, 17(1): 2853-2884.

---

> > > ### Author Response · Authors · 2019-11-14
> > > **Response to specific points from reviewer 3**
> > >
> > > We thank Reviewer 3 for taking the time to respond to us on such short notice, and provide analytic points. Some of the expressed concerns are already addressed in the paper. We apologize if some things were not stated clearly enough, and re-iterate below to avoid further confusion.
> > >
> > > 1. We agree that mtl is not limited to vision tasks alone. Indeed, it would be interesting to also consider other modalities (e.g. text) from within a single framework. However, looking at the majority of related works, we find that most frameworks have constrained themselves to a single modality too. Consider a list of papers that also considered visual tasks alone: Hand & Chellappa,'17; Liu et al.,'15; Lu et al.,'17; Kendall et al.,'18; Kokkinos et al.,'17; Misra et al.,'16; Xu et al.,'18; Sener & Koltun.,'18. etc. Similarly, some papers only considered text classification tasks: ‘Liu P. et al., '17 Adversarial MTL for Text Classification’; ‘Ruder S. et al., '17, Sluice networks';
> > >
> > > To alleviate Reviewer 3 concern, we will formulate our claims more carefully and explicitly mention (on title and text) that we were mostly interested in dense prediction tasks in this work, and as such we only tested our method on vision tasks.
> > >
> > > 2. We agree that the paper does not provide theoretical evidence for this claim, so this will be mentioned explicitly in the paper. However, the paper does provide abundant experimental support for the claim that measuring the task affinities on the feature space through RSA is better compared to Lu et al., 2017, at least when using the affinities to cluster the tasks in a MTL net.
> > > Experimentally, what Reviewer 3 asked for can be verified by deriving a branched MTL net through the exact same setting for the clustering algorithm, backbone, anchors, training, etc., with the sole differentiation being that the affinity scores are either calculated through RSA or through Lu et al,2017. This is exactly what is done for the Ours-Thin-32 and Branch-32-2.0 model in Table 3 (CelebA). Similarly, we find these results in Fig. 2b (Cityscapes) when comparing the architectures found by the RSA affinities (black) against the ones from Lu et al. (green). Table 2 (Taskonomy) also contains these results (Ours vs FA), but we did not sample all architectures exhaustively as on Cityscapes due to the larger amount of tasks. Given the above, we believe there is more than enough experimental evidence on three different datasets for our claim.
> > >
> > > 3. We agree that our work could benefit from a computational cost analysis. Reviewer 1 also pointed out this issue. For a brief analysis on the computational costs we refer to the response that we formulated to Reviewer 1 (point 2). We will include the computational analysis in the final version of the paper. Given the limited rebuttal time it is hard to provide exact timings now, but taking into account that the difference between our method and exhaustive search is at least in the order of magnitude following the response to Reviewer 1, we think this is adequate for now. If not, and timing information is still desired, we will gladly provide rough estimations.
> > >
> > > 4. We evaluated our models at the end of every epoch, as is the common practice. As mentioned in the paper, the evaluation on Cityscapes and Taskonomy was based on the mtl performance (Maninis et al., 2019). The tasks on CelebA are evaluated through the mean  accuracy. Our suppl. materials contain a detailed overview of the hyperparameters that were used in each experiment, including the number of epochs/iterations. We refer Reviewer 3 there. We also included explicit references to the suppl. material when discussing the experimental setup for every dataset. Regarding the specific issue of the small performance differences in Table 3, we ran the experiment three times and found the test performance to only differ by +-0.01%. We will include a standard dev. for our test performance on CelebA to the paper.
> > >
> > > 5. The question is whether the performance improvements of the branched networks can be attributed to the use of more layers, rather than the use of a specific task grouping. This can be verified through the results reported in the paper. Consider an experiment where we consider multiple branched networks with a specific shape, but different task groupings. By comparing the performance obtained with different groupings, we can see the impact of the used task grouping. For CelebA, this can be seen from comparing the branched-32-2.0 with Ours-32 model. For Cityscapes, the results can be seen from Fig 2b. For Taskonomy, the results can be seen from Table 2. On all three datasets, our proposed task groupings have a positive effect. Bottom line, the performance increase does not only come from the increased number of layers. Note, in Table 2,  the groupings from Lu et al., 2017 (FA) even decrease the performance compared to the MTL baseline. So, with a suboptimal grouping increasing the number of layers can have a negative effect.

---

### Official Review · AnonReviewer2 · 2019-10-22
**Official Blind Review #2**

**Rating:** 6

**Review:**

This paper presents a method to infer multi-task networks architecture, more specifically to determine which part of the network should be shared among different tasks. The main idea is to first train a specific, standalone encoder/decoder network for each task. Subsequently, a task affinity factor is computed by looking at the similarity (or, more likely the dissimilarity) of an holdout set of images feature representations. Knowing these dissimilarities (based on RDM), one can cluster the tasks and make similar tasks share more of their layers. Computational budget can also be taken into consideration in the form of the number of parameters. Results on Cityscapes, Taskonomy, and CelebA datasets show, to some extent, improvements against the state of the art.

The paper is well written and addresses a common problem in multi-task learning. The experiments provided are extensive and cover most of the current multi-task learning methods and interesting problems. I especially like the idea of formalizing the dissimilarity between tasks using RDM. There are, however, a few key points that would need improvement.

First, except for CelebA, the experiments provided use ResNet50 with only 4 different "anchor point" in the network. In other words, the task at hand is limited to selecting the best sharing point between 4 choices. This is not wrong per se, but in my opinion, it does not tackle the main problem: what to do when brute force / exhaustive exploration cannot be fulfilled? CelebA provides a more complex case, but it also requires to change the method from an exhaustive search to a beam search (end of Sec. 3.2). Doing so get us back to a kind of greedy approach, precisely what was advocated against the paragraph before (in the discussion about Lu et al. 2017).

Second, the fact that task affinities are computed a priori leads to the following conclusion: "this allows us to determine the task clustering offline". While I agree that this could be useful, one has to keep in mind that compared to other methods, this one has to first train a network for _each_ task independently, which can take a long time.

Out of curiosity, did you consider other correlation coefficients than Spearman? Why use a rank-based method?

Overall, this is a nice paper, with adequate methodology. It is not groundbreaking, and most of the good results arise when we consider both performance and number of parameters, but it is interesting for the community nonetheless. I am not sure the impact would be very high, since it can probably be replaced by an architecture exhaustive search if the number of tasks and branching points in the network are low, but the formalism of the approach is welcomed.

**Experience Assessment:**

I have published one or two papers in this area.

**Review Assessment: Checking Correctness Of Derivations And Theory:**

I assessed the sensibility of the derivations and theory.

**Review Assessment: Checking Correctness Of Experiments:**

I assessed the sensibility of the experiments.

**Review Assessment: Thoroughness In Paper Reading:**

I read the paper thoroughly.

---

> ### Author Response · Authors · 2019-11-12
> **Response to reviewer 2**
>
> We thank the reviewer for making valuable comments on our paper. We address the concerns expressed by the reviewer below.
>
> 1. This is a valid point indeed. The main reason for limiting ourselves to four anchor points in the experiments on Cityscapes and Taskonomy was that it would allow a fair comparison to prior work. First, it makes it possible to exhaustively train all possible trees on Cityscapes. This information was used to have a very thorough evaluation of the task groupings (fig. 2b). Secondly, by placing the four anchor points after every block, we were able to draw a fair comparison with cross-stitch networks and NDDR-CNNs.
>
> We expect the results to further improve when we consider more fine-grained anchor points for the following reasons.
>
> (A) As pointed out by the reviewer, we show that we can handle more anchor points in our experiment on the CelebA dataset. Indeed, we were required to change the approach to a beam search. The main reason for this was the large number of tasks, rather than the increased number of anchor points. Given that we had 40 tasks and we cluster the tasks in k groups, then the number of possible groups is 40!/(k!(40-k!)). Now we need to consider every possible value 1 < k < 40. Now this clearly becomes unwieldy, especially since we need to derive the trees by considering all layers.
>
> We agree that using a beam search procedure means that we resort back to a greedy search procedure. However, enlarging the beamwidth is considerably cheaper in terms of computational costs when compared against Fully-Adaptive Feature Sharing. In particular, a larger beamwidth means that we consider a larger set of trees for which we simply have to calculate the sum of the task affinities from multiple layers. In Fully-Adaptive Feature Sharing, a larger beamwidth means training an increased number of trees.
> Also, from what we observed in our experiments we found that there is little noise in the calculation of the task affinities. This reduces the probability that the best task grouping falls outside of the beam due to noise in the measurement of the task affinities.
>
> Bottom line, we don’t think the use of a beam search should hinder the use of more anchor points.
>
> (B) We found that the task affinities at the layers between two anchor points are usually interpolations of the affinities at the anchor points. This means that we would obtain a similar tree when repeating the experiments on Cityscapes and Taskonomy. This time, the branching point would lie somewhere in between the two anchor points that are currently surrounding it. Results are expected to be similar in terms of performance, but a more fine-grained trade-off between performance and complexity is enabled.
>
> 2. We agree to the second point that determining the task groupings requires some overhead as it requires to train a set of single task networks first. However, we also feel that such overhead is justified for multiple reasons.
> - When compared against Fully-Adaptive Feature Sharing, which determines the task groupings in an on-line manner, our method achieves better results. Also, as the clustering in Fully-Adaptive Feature Sharing happens on-line, the point in time at which this happens needs to be decided. Eventually, this means that the clustering will be dependent on what happened during training up to that point. This situation hurts repeatability.
> - Cross-stitch networks and NDDR-CNNs are composed from a set of single-task networks that are usually pre-trained on the single tasks first. This means our method does not occur any significant overhead compared to these works.
>
> 3. Regarding the use of a rank-based method for calculating the RSA matrices, we ran an extra test and verified that we got the same task groupings on Cityscapes and Taskonomy when using pearson correlation. We will add this as a remark to the paper. For a more detailed analysis on the representational similarity analysis, we refer to ‘Kriesgeskort et al., Representational similarity analysis-connecting the branches of systems, 2008’.

---

### Official Review · AnonReviewer1 · 2019-10-26
**Official Blind Review #1**

**Rating:** 1

**Review:**

This paper provides a mechanism of building multi-task shared layer model, by computing Representation similarity between different layers which in turn computes the task affinity. Tasks that have a higher similarity have more shared layers, while tasks with lesser similarity are branched earlier.

Pros:
1. The question in consideration is very important
2. Yes, it is true that there are no enough studies that has studied this problem in a principled way.
3. The results are shown in some very recent datasets including Taskonomy

Cons:
1. There is not much novelty in this work. The important/ key aspect of this paper is the RSA (Representation Similarity Analysis) or the RDM (Representation Dissimilarity Matrices). This is already proposed. The rest of the paper is mere brute force. They compute the RSA between two tasks, at multiple predetermined layers of a model and whichever layer shows a lesser RSA, they start branching there. This is mostly brute force and part heuristics

2. One of the key challenge in the literature was that NAS/. Zamir et al.(2018) was computationally expensive. Very surprising that the authors did not spend a good analysis over the computation time of the proposed approach. As far I can understand, the computation of the proposed approach is going to be really expensive.
a. Train a task individual DL model on each of the task
b. For every pair of tasks, both the pretrained individual models, at multiple pre-determined layers, we have to compute the RSA
c. Once the big RSA matrix is computed, then we need to compute the correlation.
d. Thus, compute, at which point to share between these two tasks
e. Repeat this for every pair of tasks.

This is really computationally expensive, and brute force - depending on the sampling rate of the layers

3. I really dont like the comparison made in this paper with NAS - NAS really is a search optimization problem, finding new architectures. However, the technique proposed in this paper is more of a brute force comparison at pre-determined layers to compute which is the better of determined subset of n layers. This is not like a NAS problem.

4. There is no generalization in this paper- we cannot say that for ResNet always share at this layer. For every pair of tasks, and for every given model, we need to train the individual model and find the correlation for every pair of layers and perform the whole computation again. There is no takeaway for me a researcher from this paper.


**Experience Assessment:**

I have published in this field for several years.

**Review Assessment: Checking Correctness Of Derivations And Theory:**

I carefully checked the derivations and theory.

**Review Assessment: Checking Correctness Of Experiments:**

I carefully checked the experiments.

**Review Assessment: Thoroughness In Paper Reading:**

I read the paper thoroughly.

---

> ### Author Response · Authors · 2019-11-12
> **Response to reviewer 1**
>
> We thank the reviewer for taking the time to give his/her comments on our paper. However, we find the criticism harsh (brute-force, NAS) and in certain cases unjustified (computational cost, generalizability). Let us elaborate further:
>
> 1. Indeed, we use RSA to measure the task affinity which has been proposed earlier by Dwivedi et al. (2018) for characterizing task transfer relationships. However, the application of RSA to cluster tasks in a multi-task network, as proposed here, has never been explored before. Furthermore, as pointed out in the review, the question in consideration is an important one, and as such, building efficient approaches to tackle it should be equally important research-wise. We believe we proposed an efficient, far from brute-force, approach for tackling this problem, as agreed upon by Reviewer 2. This is also supported by the state-of-the-art experimental results. In addition to the method, our paper also contains an extensive analysis on prior task clustering methods and network architectures for multi-task learning. On their own, these are arguably important, and more than relevant to the multi-task learning community.
>
> 2. Although we agree that the paper could benefit from an explicit analysis over the computation time of the proposed method, we believe that the computational costs are overestimated by Reviewer 1. A reference is made to Zamir et al. (2018), but as mentioned multiple times in the paper we calculate the task affinity scores similar to Dwivedi et al. (2018). We make a brief analysis here to identify the computational costs related to the different steps. We reuse the notation from the paper.
>
> a. Train N single task networks.
>
> b. Compute the RDM matrix for all N networks at D pre-determined layers.
> This requires to compute the features for a set of K images at the D pre-determined layers in all N networks. The K images are usually taken as held-out images from the train set. We set K = 500 in our experiments. In practice this means that computing the image features comes down to evaluating every model on K images. The computed features are stored on disk.
> The RDM matrices are calculated from the stored features.
> This requires to calculate N x D x K x K correlations between two feature vectors (can be performed in parallel). Computation time is negligible in comparison to training the single task networks.
>
> c. Calculate the RSA matrix at D locations for N tasks.
> This requires to calculate D x N x N correlations between the lower triangle part of the K x K RDM matrices. Computation time is negligible in comparison to training the single task networks.
>
> Bottom line, the computational cost of our method boils down to training N single task networks plus some overhead. This is significantly cheaper than a brute-force approach that would exhaustively train all possible trees, as Reviewer 1 suggests we are doing.
>
> 3. The problem we consider here can easily be reformulated as a NAS problem in terms of a search space, search strategy and performance estimation strategy. The search space is defined as the set of trees that can be derived from a given encoder. The main reason is that a NAS procedure which would jointly determine the layer sharing scheme and underlying network structure would be considerably more expensive, if not prohibitive. As mentioned in the paper, these methods are lacking in the literature, with the exception of a few more viable alternatives, and we invite Reviewer 1 to point us to ‘pure’ MTL NAS works in case we missed them. Furthermore, despite the fact that we focus on optimizing the layer sharing in this paper, improvements in the encoder/decoder architecture can be considered orthogonal to our approach, where traditional single-task NAS can be used. Finally, our beam search can be considered as a search strategy for finding performant MTL architectures. The search relies on the task affinities, which serve as a proxy for MTL performance. In other words, the performance estimation is performed indirectly through the task affinity scores.
>
> 4. We find the claim that there is no generalization in this paper to be gravely exaggerated. From the experiments, the method clearly generalizes to different backbones (ResNet, VGG), datasets (CelebA, Taskonomy, Cityscapes) and tasks (multi-attribute classification, multiple dense prediction tasks, scene categorization). In fact, we have tested our method under more circumstances than most prior work (Liu et al.,2019; Lu et al.,2017; Hand & Chellapa.,2017; Mistra et al, 2016; Gao et al.; 2019; Neven et al.,2017). Most importantly, Reviewer 1 claiming that there is no takeaway for a researcher here, essentially dismisses a whole line of work, like Lu et al., 2017, Hand & Chellapa, 2017, etc. that explicitly tried to tackle the same problem under this branching setup.

---

### Decision · Program_Chairs · 2019-12-19

**Decision:**

Reject

**Comment:**

The authors present an approach to multi-task learning. Reviews are mixed. The main worries seem to be computational feasibility and lack of comparison with existing work. Clearly, one advantage to Cross-stitch networks over the proposed approach is that their approach learns sharing parameters in an end-to-end fashion and scales more efficiently to more tasks. Note: The authors mention SluiceNets in their discussion, but I think it would be appropriate to directly compare against this architecture - or DARTS [https://arxiv.org/abs/1806.09055], maybe - since the offline RSA computations only seem worth it if better than *anything* you can do end-to-end. I would encourage the authors to map out this space and situate their proposed method properly in the landscape of existing work. I also think it would be interesting to think of their approach as an ensemble learning approach and look at work in this space on using correlations between representations to learn what and how to combine. Finally, some work has suggested that benefits from MTL are a result of easier optimization, e.g., [3]; if that is true, will you not potentially miss out on good task combinations with your approach?

Other related work:
[0] https://www.aclweb.org/anthology/C18-1175/
[1] https://www.aclweb.org/anthology/P19-1299/
[2] https://www.aclweb.org/anthology/N19-1355.pdf - a somewhat similar two-stage approach
[3] https://www.aclweb.org/anthology/E17-2026/